# AI-Enabled Wearable Medical Internet of Things in Healthcare System: A Survey

**Fazli Subhan** [1,2], **Alina Mirza** [3], **Mazliham Bin Mohd Su'ud** [2,*], **Muhammad Mansoor Alam** [2], **Shibli Nisar** [3], **Usman Habib** [4] **and Muhammad Zubair Iqbal** [1]

1   Department of Computer Science, National University of Modern Languages-NUML, Islamabad 44000, Pakistan
2   Faculty of Computer and Information, Multimedia University, Cyberjaya 63100, Malaysia
3   Department of Electrical Engineering, National University of Sciences and Technology, Islamabad 44000, Pakistan
4   Department of Computer Science, National University of Computer and Emerging Sciences, Islamabad 44000, Pakistan
*   Correspondence: mazliham@mmu.edu.my

**Abstract:** Technology has played a vital part in improving quality of life, especially in healthcare. Artificial intelligence (AI) and the Internet of Things (IoT) are extensively employed to link accessible medical resources and deliver dependable and effective intelligent healthcare. Body wearable devices have garnered attention as powerful devices for healthcare applications, leading to various commercially available devices for multiple purposes, including individual healthcare, activity alerts, and fitness. The paper aims to cover all the advancements made in the wearable Medical Internet of Things (IoMT) for healthcare systems, which have been scrutinized from the perceptions of their efficacy in detecting, preventing, and monitoring diseases in healthcare. The latest healthcare issues are also included, such as COVID-19 and monkeypox. This paper thoroughly discusses all the directions proposed by the researchers to improve healthcare through wearable devices and artificial intelligence. The approaches adopted by the researchers to improve the overall accuracy, efficiency, and security of the healthcare system are discussed in detail. This paper also highlights all the constraints and opportunities of developing AI enabled IoT-based healthcare systems.

**Keywords:** AI; healthcare; IoMT; wearable devices; detection; monitoring





## 1. Introduction

The needs of traditional healthcare institutions for the general public are being impacted by population expansion [1]. Medical healthcare services are neither cheap nor accessible to the general public, despite robust infrastructure and sophisticated technology. The expense of healthcare is rising globally as the world's population ages. According to the most recent data, China spent over 4634 billion yuan in 2016, accounting for around 6.36 percent of total GDP, and 7231 billion yuan in 2021, accounting for 7.1 percent of total GDP [2]. To overcome these obstacles, an intelligent healthcare system is essential to educate the public about their medical concerns and keep them up to date on their medical diseases in real time.

Users with smart healthcare can tackle various medical situations on their own. It provides for remote monitoring and tracking of the patient, which lowers the patient's healthcare expenses. It also helps clinicians offer their amenities without being constrained by location. Over the previous 10 years, the trend has turned toward smart wearable devices, smart homes, smart cities, and a functioning smart healthcare system that can help an aging person live a healthy life [3]. The Internet of Things (IoT) is a rapidly growing interactive model, where large linked devices interact to enable communication between people and objects in today's digital age [4–6]. IoT in terms of healthcare means

any device that accumulates health-related information about people via phones, smart wearable devices, bands, implanted surgical instruments, or other portable devices that can monitor health.

The IoT is used in hospitals and households to remotely monitor patients. By recognizing and treating hazardous illnesses and issues at an early stage, remote patient monitoring in the healthcare sector may dramatically minimize needless visits to medical professionals, hospital stays, re-admissions, and medical care expenditures [7]. Currently, our healthcare system is very expensive due to the patient's hospitalization until treatment duration. Smart technology that can remotely monitor patients can help solve these challenges. The Internet of Medical Things (MIoT) reduces healthcare costs by gathering and exchanging real-time patient health data with clinicians. This enables the treatment of health issues before they become major illnesses [8]. The MIoT will undoubtedly improve people's quality of life. Integrated tool development can result in a variety of useful enhancements in integrated information system services, system processing, and communications with a wide range of control. As a result, in the realm of healthcare, multiple IoT-based digital, wearable devices and applications (as depicted in Figure 1) are required to identify, track, and prevent various chronic and viral illnesses [9].

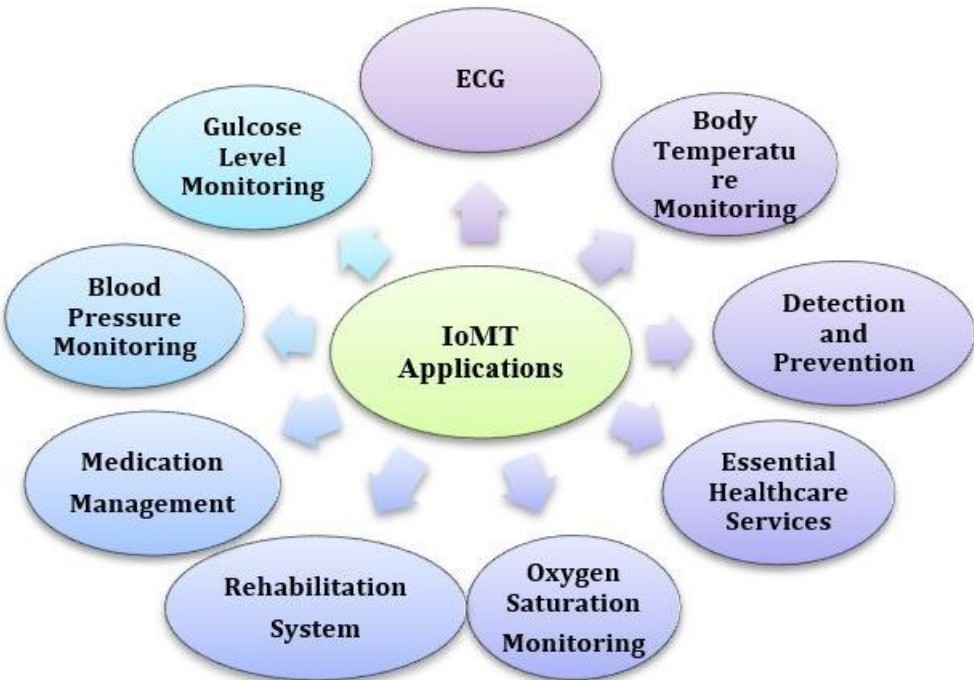

**Figure 1.** Applications of Internet of Medical Things (IoMT).

Wearable devices are one of the most important IoT technologies in the present era [10]. Wearable devices are at the center of every conversation in IoT-related healthcare systems, as they have the potential to bring about a major transformation. They are also seen to be the ideal strategy for monitoring, tracking, and detecting chronic and viral illnesses in the healthcare sector. Wearable devices, which are considered an essential aspect of the IoTs, let patients get the proper medical care at the right moment [11,12]. They have the potential to transform the standard of human existence in the current era in ways that cannot be achieved exclusively through smartphones, and customers have shown a strong desire to purchase and wear these devices. In hospitals, medical equipment along with the medical staff deployment can be tracked by IoT devices with sensors [13,14]. These devices are extensively employed in the healthcare industry to monitor patients and produce alerts for individuals who have dangerous conditions.

In the last two decades, researchers have invented various wearable devices that can monitor vital health constraints such as pulse rate, breathing, blood pressure, lungs sound,

body actions, temperature, etc., as shown in Figure 2. Due to the variety of wearable devices in the market for monitoring different health parameters, it is difficult to select the appropriate smart wearables from the market. Hence, there is a real need to investigate the existing wearable in terms of their performance and identify their discrepancies. Therefore, in this paper, the authors aimed to focus on and discuss the most trending wearables in the medical field for the purpose of detection and prevention of multiple diseases. A comparison of the most widely available wearable devices for monitoring the vital health signs of humans and infectious diseases such as COVID-19 is carried out in this paper [15–20]. Moreover, the importance of improved healthcare services due to IOT technology is highlighted. In Section 2, the different IoT healthcare applications are explored. Smart wearables for healthcare are presented in Section 3. Section 4 presents the challenges and issues related to smart wearables devices, followed by the conclusion in Section 5.

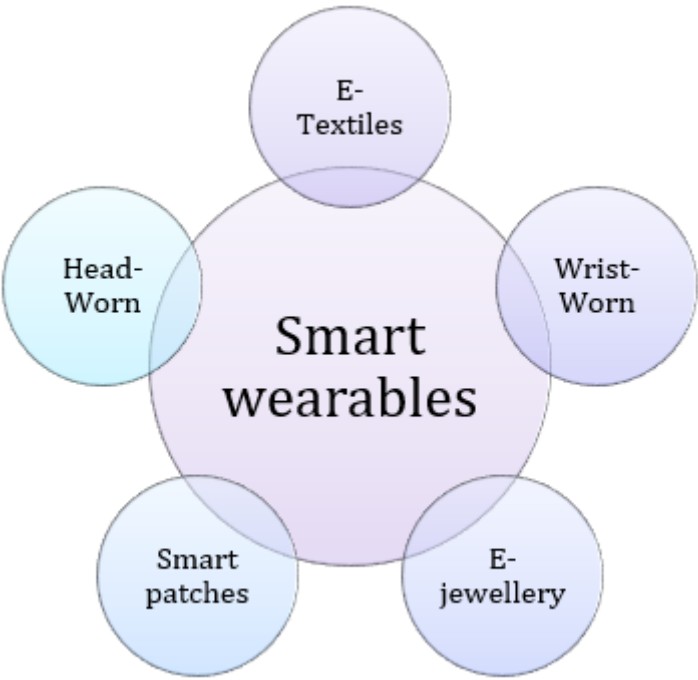

**Figure 2.** Categorization of wearable devices.

## 2. Literature Review

*Internet of Things in Healthcare*

IoT based health monitoring solutions and medical wearables are enabling the creation of hospitals "without walls", where medical healthcare professionals from different fields provide their healthcare services remotely to outpatients, thus freeing up vital bed space for patients who require extra care and minimizing the cost burden. It also allows people to have extra control over their fitness all the time. Before COVID-19, remote health monitoring systems had already gained popularity. During the COVID-19 pandemic, the demand for smart technology and wearable medical devices increased for remote patient monitoring [21]. It is assumed that demand for the devices in the market will double over the next 5 years. According to the Royal College of General Practitioners, 71 percent of regular consultations were conducted remotely in the four weeks leading up to 12 April 2020, compared to only 25% in the same period in 2019 [21]. This is a massive change.

An IoT based health supervising system has multiple advantages in the medical field, but a few drawbacks are also present: the need to regularly reconfigure individual sensors to maintain supervising process accuracy; connectivity loss in case of device battery failure, and occurrence of security hazards due to massive storage of data in a single database. IoT healthcare has been developed for several purposes, such as diabetes

sensing, BP monitoring, rehabilitation systems, electrocardiogram supervision, assisted ambient living (AAL), etc. Blood sugar monitoring shows individual patterns of change and facilitates making diet plans and prescribing medicines [22]. A system for monitoring heart activities, i.e., IoT-based ECG monitoring, was proposed in [23]. A few IoT- based healthcare applications are listed in Table 1.

**Table 1.** IoT healthcare applications and precise use cases [24].

| Applications | Details | Technology | References |
|---|---|---|---|
| Electrocardiogram (Monitoring, Detection) | A wireless procurement IoT-related ECG examining device | IoT framework Anomaly detection | [23,25] |
| Glucose Level Sensing (Monitoring) | Blood sugar level estimating noninvasive device comprising a sugar level accumulator | IoT, IPv6 | [23,25,26] |
| Body Temperature (Monitoring) | A thermometer along with an IoT channel are used in this application | Thermometer, IoT System, Sensors | [25,27,28] |
| Blood Pressure (Monitoring) | To create an IoT-based BP monitor, BP KIT meter and mobile device are interleaved | IoT frameworks, Pressure Sensor | [25,29] |
| Oxygen Saturation (Monitoring) | Real-time data transfer between patient and practitioner is done by joining oximeter | Oximeter, IoT system | [25,30,31] |
| Medicine Management (Monitoring, Detection) | A medication managing system scrutinizes machines validity using I2Pack and the iMedBox | RFID, IoT | [25,32,33] |
| Rehabilitation System (Monitoring, Detection) | A proposed IOT based system that continuously gives details of the mental state of patients | IoT, Machine Learning | [25,34,35] |
| Smart Devices (Monitoring, Detection | Data is captured, processed, and sent in real time | IoT, Big Data | [25,36,37] |
| Wheelchair Management (Monitoring, Detection) | Fully autonomous wheelchair app comprising potential wheelchair design | IoT Frameworks, Machine Learning | [25,38] |
| Vital Healthcare Services (Monitoring) | Cardiovascular diseases, aptness, and nervous | Machine Learning,Cloud Computing, Big Data | [25,39–41] |
| Imminent Healthcare (Monitoring, Prevention) | Mobilized IoT-based services, designs, and applications help practitioners | IoT Net framework, Intelligent Security Model | [40,42,43] |

In [44], an instrument for gathering and conveying blood pressure data over an IoT network was presented. Monitoring human body temperature is an important aspect of healthcare, as it is considered a dynamic sign in controlling homeostasis [27].

## 3. Wearable Technologies for Different Body Parts

Different aspects, such as optical, electrical, piezoelectric effects, and electrochemical all have a vital part in the operation of sensors in the healthcare sector. Different parts of the human body have their own wearable devices. Wearable devices are designed to provide useful information to their users while they are wearing them. These wearable devices are categorized by their position and material as shown in Figure 2.

In this section, a few of the most significant wearables worn on the human body for the purpose of healthcare monitoring are summarized.

### 3.1. Smart Thermometer

There are a variety of IOT based thermometers developed that are low cost, small in size, and wearable [10] and available in different forms, such as touch, patch, and radiometric [45]. A very popular type of thermometer was developed by a US-based company called KINSA to detect whether people had a high fever in the pandemic; this thermometer is connected to the Internet as well. This thermometer was launched for the

purpose of detecting people having symptoms of flu with fever, and the developer can now determine the most suspected sectors of the states of the USA on the basis of recorded temperature [46,47]. Another thermometer is Temp Traq, which is a wearable wireless thermometer that monitors temperature continuously for 24 h and is waterproof [48]. Tucky is also a prominent type of smart thermometer that performs well in dangerous infections and febrile seizure conditions [49]. A variety of small size smart thermometers are also available in the market, such as iSense, iFever, and Tempdrop, which are reliable and helpful for monitoring chronic and viral diseases.

### 3.2. Smart Helmet

During pandemics such as COVID-19, the wearable smart helmet has gained popularity, and it has proved to be safer and gives good results as compared to an infrared thermometer gun. This helmet is a combination of a smart helmet and a thermal camera [50]. When the individual body temperature rises, the person's location and image are captured by an optical camera, and this information is sent to the relevant authorities so that they can make timely decisions and locate the suspected individual. In this scenario, Google locates the person and enhances the performance by also locating the places visited by the person. Currently, Italy, China, the USA, and the UAE have used this wearable helmet to observe gatherings within a distance of 2 m for the purpose of crowd control during the COVID-19 pandemic [20]. Another commercially available product, the KC N901 smart helmet, has the ability to detect high temperatures in gatherings, trace 200 individuals within a minute, and achieve an accuracy of near to 96% [51].

### 3.3. Wrist Band

A wrist band is a wearable that is worn on the wrist or perhaps on the forearm. Currently, this wearable is also used by researchers for different purposes in the field of healthcare to monitor diseases. The E4 wristband is deployed for run-time streaming and investigation of physiological data. Research conducted by [52] applied the E4 wristband to detect the early symptoms of influenza and rhinovirus. The authors measured the individual movement, body temperature, daily life activity, and pulse rate with the help of the E4 wristband. It was found that this wearable wristband is simple and feasible for monitoring the individual during a viral disease spreading period, and it also makes a good contribution to limiting the spread of infections related to respiratory systems, such as during the COVID-19 pandemic. In [53], a wristband IoT based healthcare kit was designed that monitored the vital signs of a person's health such as pulse rate, temperature, BP, and sent them to the specific medical practitioner with an alert.

### 3.4. Smart Watches

According to the Garter report, smartwatches are considered the most prevalent wearable devices [54]. Based on their functionality, smartwatches are divided into two categories: first, for communication purposes such as sending messages, emails, and phone calls; and second, for healthcare related services, such as sending human biometric data to healthcare centers (e.g., pulse rate and body temperature [32]), tracking heart rate, and tracking burned calories and step count [55]. After several studies, it was found that smart watches are mainly applied in healthcare applications that are related to health monitoring or for monitoring senior citizens [56], such as fall detection, and also for chronic diseases monitoring and medication [57]. In the current period of the pandemic, these watches are used to control and prevent infectious diseases from spreading; for example, hand washing can effectively control the spread of viral diseases. Moreover, in [58], a handwashing assessment system was developed that is based on a smartwatch with Bluetooth beacons to control the spread.

### 3.5. Smart Glasses

Smart glasses are also significant IoT based wearables that play an important role in the healthcare IoT. As compared to other thermometers, these glasses do not have much interaction with humans. In these glasses, optical or thermal cameras are applied to monitor the activity of crowds [59] or to detect and trace a suspected person with high temperature, and its detection technology makes it more reliable. There are several smart glasses present in the market that can monitor different numbers of people, such as Rokid smart glasses [60], that can trace and monitor a group of 200 people. These glasses are used in combination with a camera as well. These glasses are helpful for authorities and healthcare staff during the spread of viral diseases such as Ebola, which is spread from person to person by body fluids via the eyes and other body parts [61]. Similarly, these glasses were helpful in monitoring crowds and assisting doctors and patients in order to protect them from COVID-19 and other diseases.

### 3.6. Smart Jacket

A smart jacket is also an important wearable from the intelligent garments category. It contains a combination of sensors and a wireless platform that are used to monitor oxygen saturation, breathing rate, and some other safety purposes. In [62], a smart sensor-based outdoor jacket was designed that is connected to a smartphone application system to monitor fall detection, emergency calls, heartbeat monitoring, body temperature, and breathing rate. The first COVID fight jacket available in the market is Covest [18], which has a built-in mask, sensors for social distancing, and a built-in thermometer for monitoring body temperature. Similarly, a new smart jacket is also available in the market named Mamaope [19], which monitors individuals affected by pneumonia in a more accurate way. This wearable consists of several sensors to monitor the heartbeat, body temperature, lung sounds, $O_2$ sats, and breathing. The smart jacket is also widely used in industrial environments for workers' safety. The research in [63] developed a sensor-based jacket that worked with wireless technologies to monitor the workers and hazards that occur during working and ensured safety in advance.

### 3.7. Smart Socks

The word smart socks was initially used in the literature in 2005, and it has gained popularity in smart wearable devices as a way to capture motion-related information [64]. Foot wearables are embedded with one or a combination of sensors for the purpose of collecting data and communicating between these devices for the analysis of data. Smart IOT based socks are comfortable and easy to wear [65]. Currently, these socks are widely used in healthcare and fitness-related services. According to research, every 1.2 s, diabetic foot wounds cause an infection in patients [66]. For such patients, the SirenCare Company developed smart socks that help to detect diabetic injury and control wound infection and also monitor the user movements [67]. Smart socks are also used to monitor the temperature of the human body; the authors in [68] presented a smart socks system that monitored diabetic patients' body temperature. Smart socks are also very useful for Parkinson's patients. Other commercially available smart socks are Owlet and Baby Vida, which are worn on a baby's feet to monitor the pulse rate and oxygen level with the application of Bluetooth technology [69,70].

### 3.8. Data Gloves

Gloves are the most crucial and commonly used hand wearables that provide safety to hands from different hazardous conditions. Smart gloves are armed with different sensors and technologies. The use of technology in these hand wearables is not new; the concept started approximately four decades ago [71]. The main task of these gloves is to help people manipulate and touch things in a more safe and direct way. There are different types of devices that are applied with these smart gloves that are small, lightweight, and easy to carry. Smart gloves are also applied in the healthcare sector for different purposes, such as

patient rehabilitation, vital surgery, etc. The authors in [72] presented smart gloves that were composed of electrochemical biosensors and used as screening tools and defense tools. The most important use of these gloves is for patients that are affected by joint and movement related diseases such as rheumatoid arthritis and Parkinson's. Rheumatoid arthritis is a joint disease that mainly affects the older population who need therapies and regular medication. For such types of patients [73], a smart gloves system was developed to observe and monitor the joints of such patients. During COVID-19, Keltic developed the anti-COVID touchscreen smart gloves, also known as COVID-19 gloves. According to Keltic, these touchscreen smart gloves are not only antiviral but also anti-bacterial [15,20].

### 3.9. Smart Mask

Face masks are considered as as primary measure to fight against airborne pathogens, similar to other precautions [74]. In order to avoid the spread of infectious diseases, distancing, washing hands, and quarantine are required [75]. For this purpose, the digitalization of a mask during a pandemic plays a vital role in personal safety in the present connected IOT era. In the current scenario of a pandemic, the smart mask has gained popularity. The authors in [76] present an idea to digitalize the smart mask, making it safer, improving the wearable safety to fight against infectious disease spreading from respiratory or aerosol droplets, and performing real-time data analysis to enhance the individual safety in a timely manner. The smart wearable mask in general is a combination of several technologies, such as filtering mechanisms, sensors, and material [77]. Similarly, the smart mask developed in [76] is also a mix of changeable filters, wireless sensors, rechargeable batteries, mobile applications, and real-time analysis mechanisms. A mask was developed by a research team at the University of California for the detection of proteases that are mainly enzymes and enhance the breakage of COVID-19 virus proteins [78]. A variety of smart masks are available in the market, such as Xiaomi Purely Mask, G-Volt mask, and Pure care smart mask; among all of these, the G-Volt mask gives the best filtration efficiency, at 99%.

### 3.10. Smart Stethoscope

A stethoscope is a medical device that is applied to the human body for the purpose of auscultation and transmits these data to the healthcare provider. The stethoscope is utilized to trace out a variety of health-related issues, such as flu, heart beat, stomach sounds, and many more. For this technology, the correct position for application to the human body is very important. For this purpose, the authors in [79] developed a smart stethoscope that only gives sounds when it is applied correctly on the body. Another smart wearable stethoscope, designed in [80], is a combination of sensors, with a Littmann 3200 stethoscope, embedded in a vest and placed on the human torso to hear breathing and cough sounds, and it was very helpful during COVID-19 type pandemics in which human respiratory system monitoring is a major issue. Lungs auscultation is a very crucial task during a respiratory-type pandemic. Similarly, in [81], a smart stethoscope system takes and merges different sensing parameters, such as sense background noise, breathing rate, ECG, and individual actigraphy. This wearable stethoscope patch is very helpful to perform auscultation without any gaps, and it does not need any type of sensors on different body parts for the purpose of auscultation. Another smart AI-based smart stethoscope developed by the Johns Hopkins University team that listens to lung sounds has the ability to detect pneumonia. A large variety of smart stethoscopes are available on the market, such as RESP, Stemoscope [82].

### 3.11. Smart Ring

The smart ring is a digital wearable that was first launched in 2013 [83] and is used with integrated functions of mobile phones, similar to a common wearable ring, or it may be large in size. It is used for several purposes such as activity monitoring, payments, home appliances control, and making phone calls and also plays a significant role in the healthcare sector by monitoring vital health signs, such as pulse rate, body temperature,

respiration rate [84], sleep, movements, oxygen level, and many more. A variety of smart rings are available on the market, such as the Oura Ring, which monitors individual sleep and helps the wearer to make appropriate adjustments to sleep. Oura is a wearable metallic ring, a combination of very small sensors, used to observe health-related information such as breathing, temperature, and pulse rate [85]. For the most part, such parameters are observed for viral diseases such as cold, flu, and COVID-19. The first outbreak of SARS was in 2003, MERS was in 2012, and SARS-Cov-2 is the most recent viral disease wave, with worldwide approximately 340 million cases and 5.57 million deaths [86]. For such types of parameters as cited above, the Oura ring has the potential to trace out symptoms of COVID-19 at the home level and detect common symptoms of COVID-19 virus with an accuracy of near to 90%. It has a battery timing of 1 week with a one-time charging of 20–28 min [85].

### 3.12. Smart Belt

One more generally used wearable sensor and IOT based medical device [87] for the healthcare sector is the smart belt. It is applied to different body parts for several monitoring purposes, such as for movement, ECG, body temperature, respiratory rate, heartbeat [87,88], breathing rate, fall detection, sleep, body parts postures, distance, and many more. The authors in [89] proposed a smart belt-based system to estimate social distance and temperature in order to mitigate COVID-19 effects and reduce the spread from person to person by keeping a social distance of 1 m. When an individual is near to 1 m or less, the smart belt generates an alarm and sends the distance and temperature of the nearer body to another system via Blynk software with help of IoT. These cost-effective sensor-based belt help to reduce COVID-19 spread. Similarly, other work done in [90] helps to monitor the heartbeat and respiration rate while the user is sitting, standing, or walking.

### 3.13. Smart Patches

Smart patches, or E-patches, are wearable sensors that are commonly used in healthcare and applied to the patient's skin [91]. These are tiny in size, lightweight, and can monitor the pulse rate, respiration rate, oxygen level, body temperature, and ECG [92]. The authors in [93] presented a smart patch that is a combination of different wireless, self-powered technologies for the purpose of recording body temperature and activities. Another smart patch was developed in [94] that monitors blood pressure by applying ECG and FPS sensors, giving real-time results. Similarly, the wearable patch is also used for glucose sampling purposes. During a pandemic, the main symptoms are high temperature, BP, respiratory rate, pulse rate, and breathing, among others. In order to monitor symptoms of COVID-19, a variety of patches are available on the market. Examples include the RespiraSense patch that monitors respiratory rate; the Life Signals patch that detects parameters such as respiratory rate, heart beat, temperature, and ECG; and the VitalConnect patch that can detect falling pulse rate, respiratory rate, temperature, and body activities [92].

### 3.14. Smart Contact Lens

Soft or smart contact lenses are the most prominent eye wearables that help to monitor the eyes' physiological functions. In recent years, these eye wearables have gained great attention in the healthcare sector for monitoring purposes commercially [95,96]. The first smart contact lens named Triggerfish was commercially available in 2016 for the purpose of monitoring glaucoma patients continuously for 24 h [96]. Various soft contact lenses are available that work on the base of optical or electrical mechanisms, embedded with sensors to observe eye fluids. Other work [97] helped to continuously monitor glucose and diabetic retinopathy. The smart lens is also used to provide continuous delivery of drugs in the eyes for glaucoma patients; for example, the Leo Lens smart drug delivery system for eyes. Smart lenses are also applied to overcome and decrease eye allergies [98].

### 3.15. Smart Nutrition System

Wearable technology is also used to monitor diabetic patients' dietary intake. A piezo-electric necklace nutrition monitoring system uses an algorithm to distinguish between liquid and solid intake and identify swallows [11]. Based on the outcome, this device can distinguish between solid and liquid meals. In [99], a new method for predicting future glucose concentration levels is described, employing data from a continuous glucose monitoring (CGM) device and a recurrent neural network (RNN). Using these predicted glucose levels, early hypoglycemia/hyperglycemia warnings can be issued to establish acceptable dosages of insulin.

### 3.16. Smart Mood Monitoring System

A smart monitoring device can be utilized to keep track of a patient's emotional condition. Moreover, it assists health specialists to treat mental ailments such as depression, bipolar disorder, etc. According to [100], a CNN network is used to analyze six different person's moods. Other research [101] demonstrated the significance of happiness in making decisions and helped policymakers discover the critical elements that define a person's pleasure. With the integration of advanced and machine learning systems, stress may now be identified in advance using heart rate, and the system can determine patient's stress level [102].

### 3.17. Smart Rehabilitation System

IoT devices also play a vital role in the rehabilitation of patients. In [103], a clever framework called Smart Pants was proposed for the home rehabilitation of post-stroke patients. In order to determine how the patient's weight was distributed, four biosensors were fastened to the patient's body with plastic bands, and a pressure sensor was fastened to the patient's feet. With the help of this sensor's arrangement, patients' correct posture for exercises was supervised and alerts were generated if necessary. Five of the most popular types of exercise for stroke survivors can be measured and processed by the suggested approach. The framework can then use ML techniques such as random forest, random tree, naive Bayes, and multilayer perceptron to categorize the 64 attributes that were gathered from the sensors.

### 3.18. COVID-19 Detection System

With the use of IoT devices, difficult circumstances can be managed and controlled, such as the one in 2020 when the coronavirus epidemic (COVID-19) swept the globe [104]. The wearable sensor connected to an edge node in the IoT cloud to determine the condition of the body's health. The suggested system is constructed with three layered functionalities: an Android web layer for mobile devices, a cloud layer using application peripheral interface (API), and a layer of wearable IoTs sensors. Data are processed for determining symptoms from the IoT sensor layer. For precautions, warnings, and immediate answers, the information is kept in the cloud database. Finally, the patient's family is informed through alarms sent by the Android mobile application layer [18,19,47]. The brief comparison of wearable devices for monitoring, detection, and prevention of diseases and products is shown in Table 2.

**Table 2.** Comparison of wearable devices.

| Wearable Device | Monitoring Parameters | Application Position | References |
|---|---|---|---|
| Smart Thermometer (Monitoring) | Fever Monitoring, COVID-19 | Armpit, Chest, Ear | [48,95] |
| Smart Helmet (Monitoring, Detection) | Body temperature, imaging, location monitoring, crowds monitoring, head safety | Head | [50,51] |
| Smart Watch (Monitoring, Detection) | Body temperature, fall detection, chronic disease, pulse rate, viral diseases fitness | Wrist | [57,58,105] |
| Smart Glasses (Monitoring, Prevention) | Body temperature, crowd monitoring, Ebola, COVID-19 location tracking | Eyes, Head | [60,106] |
| Smart Jacket (Monitoring) | Pneumonia, body temperature, lungs sounds, oxygen saturation, safety of workers | Chest, Arm | [107] |
| Smart Socks (Monitoring) | Diabetic infection, injury, pulse rate, Parkinson's, oxygen saturation | Feet | [69,70] |
| Data Gloves (Monitoring, Prevention) | Rheumatoid arthritis, Parkinson's, COVID-19 | Fingers, Hands | [72] |
| Smart Mask (Monitoring, Detection) | Infections, pandemics, nursing flu, influenza, COVID-19, respiratory virus | Face, Nose | [76,78,78] |
| Smart Stethoscope (Monitoring, Detection) | Heart beat sounds, stomach sounds pneumonia, breathing, influenza, flu | Chest, Ears, Neck | [80] |
| Smart Ring (Monitoring, Detection) | Pulse rate, body temperature, respiration rate, movements and gesture, oxygen level | Finger | [108] |
| Smart Belt (Monitoring, Detection) | Monitor movements, ECG, body temperature, respiratory rate, heartbeat, breathing rate | Waist, Chest | [87,88] |
| Smart Patches (Monitoring, Detection) | COVID-19, pulse rate, respiration rate, oxygen level, body temperature | Skin | [92] |
| Smart Lens (Monitoring) | Vision enhancement, allergies, glaucoma | Eyes | [109] |
| Smart Nutrition System (Monitoring, Detection) | Detect glucose concentration position sensor | Neck | [11,99] |
| Smart Mood Monitoring System (Monitoring) | Monitor mood, pulse rate | Wrist | [100–102] |
| Smart Rehabilitation System (Monitoring, Detection) | Exercise monitoring system | Skin | [103] |
| Smart COVID-19 Detection System (Detection) | Cough monitoring, GPS monitoring | Neck | [18,19,47,104] |

## 4. Issues and Challenges in Medical Wearable Devices

Issues and Challenges in Medical Wearable Devices. These advancements have a tremendous impact on ordinary people's lives and habits [110]. As the popularity of these devices increases day by day and becomes a part of daily life, there arise some issues and challenges that need to be discussed here before investing or purchasing.

### 4.1. Environment Condition

Extreme temperature and humidity values should not affect the wearable device's performance, function, and durability [111]. For example, the device should be waterproof and also respond to sunshine or rainfall. In the design and development phase, perspiration of users should be considered.

### 4.2. Comfort

The presence of the wearable device on the user's body should never be felt. The device should conform only those body parts where it is planted and should be flexible enough to allow the user to move freely.

### 4.3. Self-Governing

The device modules should work independently without human participation. These intelligent devices should be able to make their own decisions and are linked to the internet.

### 4.4. Compact

To make the wearable smart device easier to carry, it should be made as small as possible. The device should not be heavyweight and inconvenient for the user.

### 4.5. Ergonomics

While making the design of smart wearables, their physical appearance and their connection with the human body in motion must be considered carefully. The device should not obstruct the movement of the body. Headsets, finger rings, glasses, arm bands, fabrics, wrist bands, and smart patches are all worn on the body.

### 4.6. Size/Weight

Wearable devices should be small, light-weight, optimized, and reliable [32]. The weight should not exceed more than a few grams. Moreover, flexibility issues in units should be considered so that they can easily bend. For this purpose, flexible printed circuit board (PCB) designs with all of the circuitry well integrated into the PCB have been adopted. PCBs that are flexible exhibit good robustness and are heat-resistant.

### 4.7. Attachment to Body

This refers to the unit's attachment to the body. It should not hurt the person or cause any adverse effects. For example, smart devices should not be detached in case of accidental impacts. The use of biocompatible adhesives that do not irritate the user is required. Biocompatibility is required for every piece of medical equipment that comes into contact with the body. Biocompatible materials must be stable in a physiological environment and show no signs of degradation. They must be coated with non-toxic substances to behave as an insulator. In order to prevent tissue and nerve damage, materials must have smooth surfaces. They should be flexible but not stiff for easy adhesion to soft body tissues. Surgical steel, gold, some types of ceramics, glasses, and molding plastics are considered biocompatible materials. In addition, implanted devices should be hermetically sealed to prevent the entrance of body liquids and the escape of chemicals [112]. In addition, biocompatible materials must be used for the device's outer shell. Biocompatibility is required to safeguard both the device and the patient.

### 4.8. Device and Body Safety

The user should not be harmed by the device nor should the device itself be damaged. The system must have built-in safety procedures. Some of the potential risks associated with wearable technology are discussed in Table 3. Although smart wearable technology-based devices have the potential to significantly improve a user's lifestyle, they also carry a number of hazards.

**Table 3.** Potential risk associated with wearable technology.

| Wearable Device | Possible Risks |
| --- | --- |
| Burning | Sometimes during usage these wearables cause burns on human body parts such as on skin due to the high temperature of batteries. |
| Electric shock | Wearables have direct contact with the user's body or are sometimes planted in the clothes; in such cases, a minor electric shock brings a great risk. |
| Fire, Explosion | Battery explosion causes fire and high temperature that damage the skin. |
| Skin damage | Some wearables cause cuts, scratches, and wounds on human body parts, such as wearable masks that caused skin damage during COVID-19. |
| Reactions | Sometimes wearable cause chemical reactions, such as when chemicals in the fibers or metals that have contact with skin cause rashes. |

*4.9. Accessibility*

The device should be accessible to both users and healthcare professionals for monitoring, maintenance, and data readings. This also refers to the device's ability to get data from the body with ease.

*4.10. Sensory Interaction*

Sensors should be simple and easy to use and apply [113].

*4.11. Heat Effects*

The device may be damaged if it becomes too hot. As a result, the device must be protected from excessive heat. Safety measures shall be noted in the way so that the user can be safe from any type of harm.

*4.12. Reliability*

Malfunctions and false alarms are important problems that must be avoided in the long term [114]. Before the device is put to use, endurance tests must be performed. Medical devices, especially those that are life-sustaining, must be extremely reliable. The product's life can be extended to almost 5 years. However, implantable devices cannot be repaired but replaced in case of failure. With more patient wearable devices, the requirement for device robustness is growing. Many of these devices are used continuously, 24/7. Shock, vibration, dropping, overuse, perspiration, filth, pollutants, and water are among the factors that they face (shower, bath, or pool). As these medical devices are mostly used by patients themselves, the operational commands must be simple and non-technical for patients to operate the products safely and successfully. The mechanism itself must be simple to use, similar to child-resistant closures on pharmaceutical bottles. Patients with limited hand, eyesight, or mental capacity must be able to operate the device independently.

*4.13. Side Effects*

The human organs shall be safe and secured from wearable devices, which means there should be no side effects. In the long term, the device shall not harm the users.

*4.14. High Power Consumption*

Almost all of these devices run on batteries. Therefore, it is better to use small rechargeable battery cells in a device, and it must have wireless connectivity. Low-power CMOS and high-efficiency mechanical pumps can help reduce power consumption by continuously keeping equipment in sleep mode.

*4.15. Wearable/Implantable*

Batteries are commonly used as the primary power source for wearable or implanted medical devices. The cost of wearable devices depends on battery life. Although rechargeable batteries are available for wearable devices, implanted devices rely on batteries as

their primary source of power, and their replacement is done through a surgical procedure. However, an alternative approach is inductive charging, but the user/patient needs to be in contact with a charging device at all times, which is itself dangerous. Normally, a rechargeable battery must last for at least 8 h of normal use before needing to be recharged [115].

### 4.16. Normalization

There is a very large number of manufacturers working on a wide range of wearable devices. These manufacturers have set their own set of rules and designs for a device. As a result, they are incompatible with one another. This is a serious issue that requires attention. All devices should be normalized or standardized.

### 4.17. Information Privacy and Security

Almost all wearable devices, such as Bluetooth, Zigbee, and NFC, function in a network. These networks are vulnerable to cyber-attacks when connected normally. These wearable devices do not provide security for personal information such as identification and health status [116]. As a result, a secure technique must be built on these devices and networks in order to protect any information.

### 5. Conclusions

The implementation of Internet of Things technology in healthcare systems is one viable technique for alleviating the aforementioned challenges. The Medical Internet of Things is predicted to bridge many technologies in the next years to enable new applications by linking physical items to enhance intelligent decision-making in the medical sector. The IoT has the ability to reduce the burden on sanitary systems while also offering customized health services to expand people's quality of life. Smart medical device-based health management systems have the potential to improve healthcare's overall performance. Therefore, in this paper, the most trending IoT applications and medical wearables utilized in the modern era for the detection and prevention of multiple diseases are discussed and compared. Moreover, the authors have thoroughly elaborated on the challenges associated with these medical wearables that must be considered for designing better wearable devices and achieving improved healthcare services in the future. The paper has also thoroughly covered the latest diseases such as COVID-19 and monkeypox.

**Author Contributions:** Conceptualization, A.M. and S.N.; Methodology, U.H. and M.Z.I.; Software, S.N. and F.S.; Validation, U.H. and S.N.; Formal analysis, A.M. and S.N.; Case studies, M.B.M.S. and A.M.; Resources, M.Z.I. and M.B.M.S.; Data curation, M.M.A. and F.S.; Writing—original draft preparation, F.S. and A.M.; Writing—review and editing, S.N. and M.Z.I.; Visualization, M.B.M.S.; Supervision, M.M.A. All authors have read and agreed to the published version of the manuscript.

**Funding:** This research received no external funding.

**Institutional Review Board Statement:** Not applicable.

**Informed Consent Statement:** Not applicable.

**Data Availability Statement:** Not applicable.

**Conflicts of Interest:** The authors declare no conflict of interest.

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
