# Peer review of "AI-Enabled Wearable Medical Internet of Things in Healthcare System: A Survey"

_applsci, doi:10.3390/app13031394_

Round 1

Reviewer 1 Report

The authors of this article have presented a detailed survey on AI-enabled wearable medical internet of things in healthcare. The paper aims to cover all the advancement made in wearable Medical Internet of Things (IoMTs) for healthcare systems, which has been scrutinized from the perceptions of their efficacy in detecting, preventing, and monitoring diseases in healthcare. This paper thoroughly discusses all the directions proposed by the researchers to improve healthcare through wearable devices and artificial intelligence. The approaches adopted by the researchers to improve the overall accuracy, efficiency, and security of the healthcare system are discussed in detail. This paper also highlights all the constraints and opportunities of developing AI enabled IoT-based healthcare systems. Overall, the quality of the paper is very good. I would suggest adding the points given below to further improve the quality of the manuscript:

Comment 1: The article contains few grammatical mistakes. It is advised to thoroughly check to avoid grammatical or typo errors.

Comment 2: It is advised to add the latest diseases healthcare systems detail as well such as COVID-19, Moneypox etc.

Comment 3: Applications of the Internet of Medical Things (IoMTs) should be discussed in detail.

Author Response

We would like to thank the reviewers for taking out time of their busy schedules to review our manuscript in detail. We have followed up on the comments received from the esteem reviews and revised our manuscript accordingly. The new version takes into account individual comments by each reviewer. The proposed suggestions by respected reviewers have been incorporated into the revised version, which has further improved the understandability and clarity of the manuscript. In this document, we are reproducing the reviewer’s comments that we have received. Our response is included in the revised manuscript.

Reviewer 1

The authors of this article have presented a detailed survey on AI-enabled wearable medical internet of things in healthcare. The paper aims to cover all the advancement made in wearable Medical Internet of Things (IoMTs) for healthcare systems, which has been scrutinized from the perceptions of their efficacy in detecting, preventing, and monitoring diseases in healthcare. This paper thoroughly discusses all the directions proposed by the researchers to improve healthcare through wearable devices and artificial intelligence. The approaches adopted by the researchers to improve the overall accuracy, efficiency, and security of the healthcare system are discussed in detail. This paper also highlights all the constraints and opportunities of developing AI enabled IoT-based healthcare systems. Overall, the quality of the paper is very good. I would suggest adding the points given below to further improve the quality of the manuscript:

Comment 1: The article contains few grammatical mistakes. It is advised to thoroughly check to avoid grammatical or typo errors.

Our response: Thank you for the comment. The entire paper is thoroughly revised again as suggested by the respected reviewer. All the typos errors and grammatical mistakes are removed in the revised manuscript. 

Comment 2: It is advised to add the latest diseases healthcare systems detail as well such as COVID-19, Moneypox etc.

Our response: Thank you for the comment. The latest diseases’ healthcare systems are added in the revised manuscript. The authors have added several healthcare systems of the latest diseases in general and COVID-19 & monkeypox diseases in particular. 

 Comment 3: Applications of the Internet of Medical Things (IoMTs) should be discussed in detail.

Our response: Thank you for the comment. A number of applications of the Internet of Medical Things (IoMTs) are discussed in detail. Furthermore, one detailed table is added in the revised manuscript about IoMTs to further improve the clarity and understandability of the revised manuscript.

Author Response

We would like to thank the reviewers for taking out time of their busy schedules to review our manuscript in detail. We have followed up on the comments received from the esteem reviews and revised our manuscript accordingly. The new version takes into account individual comments by each reviewer. The proposed suggestions by respected reviewers have been incorporated into the revised version, which has further improved the understandability and clarity of the manuscript. In this document, we are reproducing the reviewer’s comments that we have received. Our response is included in the revised manuscript.

Reviewer 2

Artificial Intelligence (AI) and the Internet of Things (IoT) are extensively employed to link accessible medical resources, deliver dependable, and effective intelligent healthcare. Body wearable gadgets have currently flickered attention as powerful devices for healthcare applications. This study provides a survey of the advancement made in wearable Medical Internet of Things (IoMTs) for healthcare systems. Therefore, I think that the paper makes a contribution and has the potential to be published in the Applied Sciences. However, I summarize in the GENERAL COMMENTS as follows: 

Comment 1: The novelty of this manuscript is limited. The authors should clearly explain the importance and value of this survey.

Our response: Thank you for the comment. In this survey, the authors have conducted a thorough examination of wearable Internet of Things devices that have been used to track health throughout the preceding 20 years. Modern wearable technology, such as ECG monitoring, glucose sensing, blood pressure monitoring, and wheelchair management Covid 19 symptoms detecting gadgets that are used today, is detailed in depth. People can stay active and get emotional fulfillment through technology thanks to wearable technologies.

Comment 2: Section 3 presents the smart wearable devices for healthcare, and Section 4 presents the challenges and issues related to smart wearable devices. These two sections are very important for the full paper. There are many such applications of wearable devices, and the author should give an overview by category. Unfortunately, the author is just a simple literature list. The author should give an in-depth analysis.

Our response: Thank you for the valuable comment. According to the suggestion, we have made appropriate changes in section 3. Subsections (3.15–3.18) are included that added new diverse forms of wearable technology.

Reviewer 3 Report

Abstract :  Please specify the main research of the paper

Introduction :  Please add more reference that give a novelty of the manuscript

Table 1, 2 and 3 are too small, please enlarge it

Conclusion :  Please add more conclusive result

Author Response

We would like to thank the reviewers for taking out time of their busy schedules to review our manuscript in detail. We have followed up on the comments received from the esteem reviews and revised our manuscript accordingly. The new version takes into account individual comments by each reviewer. The proposed suggestions by respected reviewers have been incorporated into the revised version, which has further improved the understandability and clarity of the manuscript. In this document, we are reproducing the reviewer’s comments that we have received. Our response is included in the revised manuscript.

Reviewer 3

Comment 1: Abstract :  Please specify the main research of the paper.

Our response: Thank you for the comment. The abstract is revised again to specify the main research of the paper. The paper aims to cover all the advancement made in wearable Medical Internet of Things (IoMTs) for healthcare systems, which has been scrutinized from the perceptions of their efficacy in detecting, preventing, and monitoring diseases in healthcare. The latest healthcare systems are also included such as COVID-19 and monkeypox. This paper thoroughly discusses all the directions proposed by the researchers to improve healthcare through wearable devices and artificial intelligence. The approaches adopted by the researchers to improve the overall accuracy, efficiency, and security of the healthcare system are discussed in detail. This paper also highlights all the constraints and opportunities of developing AI enabled IoT-based healthcare systems.

Comment 2: Introduction :  Please add more reference that give a novelty of the manuscript.

Our response: Thank you for the comment. The latest references are added in the revised manuscript to further improve the understandability of the manuscript.     

Comment 3: Table 1, 2 and 3 are too small, please enlarge it

Our response: Thank you for the comment. Tables 1, 2, and 3 are replotted to improve their quality and make them more visible.

Comment 4: Conclusion :  Please add more conclusive result

Our response: Thank you for the comment. The conclusive results are revised again with more valuable remarks. This is highlighted in the revised manuscript.

Round 2

Reviewer 2 Report

Referee report

Artificial Intelligence (AI) and the Internet of Things (IoT) are extensively employed to link accessible medical resources, deliver dependable, and effective intelligent healthcare. Body wearable gadgets have currently flickered attention as powerful devices for healthcare applications. This study provides a survey of the advancement made in wearable Medical Internet of Things (IoMTs) for healthcare systems. The authors carefully revised the manuscript and made some changes to the version according to the comments of the reviews. Therefore, I think that the paper makes a contribution and has the potential to be published in the Applied Sciences.

Author Response

Dear Editor,

We would like to thank the reviewers for taking out time of their busy schedules to review our manuscript in detail. We have followed up on the comments received from the esteem reviews and revised our manuscript accordingly. The new version takes into account individual comments by each reviewer. The proposed suggestions by respected reviewers have been incorporated into the revised version, which has further improved the understandability and clarity of the manuscript. In this document, we are reproducing the reviewer’s comments that we have received. Our response is included in the revised manuscript.

Regards,

The authors

Reviewer 2

Artificial Intelligence (AI) and the Internet of Things (IoT) are extensively employed to link accessible medical resources, deliver dependable, and effective intelligent healthcare. Body wearable gadgets have currently flickered attention as powerful devices for healthcare applications. This study provides a survey of the advancement made in wearable Medical Internet of Things (IoMTs) for healthcare systems. The authors carefully revised the manuscript and made some changes to the version according to the comments of the reviews. Therefore, I think that the paper makes a contribution and has the potential to be published in the Applied Sciences. 

Our response: Thank you so much for appreciating our work.

Reviewer 3 Report

The revised manuscrit has been improved yet.  Please check about the reference citation style.

Author Response

Dear Editor,

We would like to thank the reviewers for taking out time of their busy schedules to review our manuscript in detail. We have followed up on the comments received from the esteem reviews and revised our manuscript accordingly. The new version takes into account individual comments by each reviewer. The proposed suggestions by respected reviewers have been incorporated into the revised version, which has further improved the understandability and clarity of the manuscript. In this document, we are reproducing the reviewer’s comments that we have received. Our response is included in the revised manuscript.

Regards,

The authors

Reviewer 3

The revised manuscript has been improved yet.  Please check about the reference citation style.

Our response: Thank you for appreciating our work. The references are cited according to the journal format as suggested by the repeatable reviewer.
